

# Traumatic surfing injuries in New Zealand: a descriptive epidemiology study

James Furness[1], Katherine McArthur[1], Debbie Remnant[2], Darcy Jorgensen[1], Catherine J. Bacon[2,3], Robert W. Moran[2], Wayne Hing[1] and Mike Climstein[1,4,5]

[1] Faculty of Health Sciences/Water based Research Unit, Bond University, Gold Coast, QLD, Australia
[2] Osteopathy, Unitec Institute of Technology, Auckland, New Zealand
[3] Faculty of Medical and Health Sciences, University of Auckland, Auckland, New Zealand
[4] Physical Activity, Lifestyle, Ageing and Wellbeing Faculty Research Group/ Faculty of Health Sciences, University of Sydney, Sydney, NSW, Australia
[5] Faculty of Health, Southern Cross University, Gold Coast, QLD, Australia

Corresponding author
James Furness, jfurness@bond.edu.au

## ABSTRACT

**Background.** New Zealand (NZ) has nearly 14,000 km of coastline and a surfing population of approximately 315,000 surfers. Given its popularity, surfing has a high frequency of injury claims, however, there remains a lack of data on traumatic surfing-related injuries from large population studies. The primary purpose of this study was to examine traumatic surfing injuries in NZ specific to injury incidence, duration, location, type, mechanism of injury and associated risk factors.

**Methods.** A sample of self-identified surfers currently living in NZ participated in an online retrospective cross-sectional survey between December 2015 and July 2016. Demographic and surfing injury data were collected and analysed.

**Results.** The survey yielded 1,473 respondents (18.3% female); a total of 502 surfers reported 702 major traumatic injuries with an overall incidence proportion of 0.34 (95% CI [0.32–0.37]). When comparing the number of injured surfers who sustained an injury at various body locations, a significantly higher proportion of competitive surfers, compared to recreational surfers, had an injury at the neck (6.8% $vs$ 4%, $\chi^2$ (1,1473) = 5.84, $P = 0.019$); shoulder (7.4% $vs$ 4.3%, $\chi^2$ (1,1473) = 6.34, $P = 0.017$), upper back (1% $vs$ 2.4%, $\chi^2$ (1,1473) = 4.77, $P = 0.043$), lower back (7% $vs$ 3.1%, $\chi^2$ (1,1473) = 11.98, $P = 0.001$) and knee (7% $vs$ 3.4%, $\chi^2$ (1,1473) = 9.67, $P = 0.003$). A significantly higher proportion of surfers who performed aerial manoeuvres compared to those who did not reported a higher proportion of knee injuries (9.7% $vs$ 3.9%, $\chi^2$ (1,1473) = 13.00, $P = 0.001$). With respect to injury duration, the shoulder represented the largest proportion of chronic injuries (>3 months) (44.4%), and the head and face represented the largest proportion of acute injuries (≤3 months) (88%). Muscle and tendon injuries were reported most frequently (25.6%) and direct contact injuries accounted for 58.1% of all injury mechanisms. Key risk factors for traumatic injury included: competitive compared to recreational status (41.0% $vs$ 30.1%, Relative Risk (RR) = 1.36, $P < 0.001$), ability to perform aerial manoeuvres (48.1% $vs$ 31.8%, RR = 1.51, $P < 0.001$) and intermediate or above skill level surfers compared to beginner surfers (35.8% $vs$ 22.7%, RR = 1.58, $P < 0.001$).

**Conclusion.** One third of recreational surfers sustained a major traumatic injury in the previous 12 months. For competitive or aerialist surfers the risk was greater, with

this proportion approximately half. Overall, the head/face was the most common location of traumatic injury, with competitive surfers being more likely to sustain a neck, shoulder, lower back, and knee injury compared to recreational surfers. The shoulder was associated with the highest proportion of injuries of chronic duration. Future research should investigate injury mechanisms and causation using prospective injury monitoring to better underpin targeted injury prevention programs.

## INTRODUCTION

Surfing is a very popular aquatic sport and activity with an estimated 37 million surfers worldwide (*Remnant et al., 2020*), and with surfing's recent qualification into the Tokyo Olympics (*International Surfing Association, 2018*) its popularity continues to grow (*McArthur et al., 2020*). The growth in popularity of surfing has also seen a growth in scientific research. A recently published systematic review by *McArthur et al. (2020)* revealed a total of 19 descriptive epidemiology studies specific to acute surfing injury. The location of these studies were in a range of countries including Brazil, United Kingdom, United States of America, Japan, Portugal and Norway, with a number of the included studies based out of Australia (*Burgess, Swain & Lystad, 2018*; *Dimmick et al., 2018*; *Furness et al., 2015*; *Lowdon, Pateman & Pitman, 1983*; *Lowdon et al., 1987*; *Taylor et al., 2004*); however, no study was identified from New Zealand (NZ) despite the popularity of the sport.

The geography of NZ includes 14,000 kilometres of coastline, varied seafloors (sand, rock, gravel, and reef), swell provided from two oceans (the Tasman Sea and South Pacific Ocean), and varied beach exposure (spits, inlets, headlands and fiords) (*Moran & Webber, 2013*). Given this unique geography surfing is popular in NZ, with an estimated 315,000 surfers over the age of 15 within a population of 3.9 million adults (*Remnant et al., 2020*; *Simas et al., 2019*; *Statistics New Zealand, 2018*). Surfing is not only a recreational pastime and competitive outlet for NZ residents but also popular with Māori, the indigenous people of NZ; with reports of surfing by Māori several hundred years ago prior to European settlement (*Te Kanawa, 2017*). Given the broad participation base of surfing within NZ, a large cross-sectional study of surfing injuries is needed to provide unique and important information about the type of injuries that occur across the spectrum of recreational and competitive surfers. Understanding surfing aetiology in NZ, where the sport is already very popular, may reflect the range of injuries that will be sustained globally as surfing increases in popularity globally.

To the authors knowledge, only one study has investigated acute surfing injuries in NZ. *Moran & Webber (2013)* investigated surfing related injuries requiring first aid on NZ beaches. This study, however, included body surfing and body boarding and therefore limits its applicability to a surfing cohort. Furthermore, no musculoskeletal injuries such

as joint sprains or muscular strains were represented in the results, despite this injury type being widely represented in previous surf injury related research (*Furness et al., 2015*).

Other studies have explored beach and adventure sport related injuries occurring in NZ, however their primary aims were to analyse the number and cost of claims, prevalence of injury and risk of participation in various sports that result in injury (*Bentley & Page, 2008*; *Bentley, Page & Macky, 2007*). While these studies fail to provide detailed information specific to injury type, mechanism and individual surfing related risk factors; they clearly show that surfing is one of the top three adventure sports with the highest injury claim counts within NZ (*Bentley, Page & Macky, 2007*).

Previous research has investigated acute (*Furness et al., 2015*) or chronic (*Furness et al., 2014*; *Nathanson, Haynes & Galanis, 2002*) surfing related injuries and definitions of these terms have encompassed both the onset and duration of the injury. However, *Verhagen & Van Mechelen (2010)* recommend that appropriate classification of terms for injury onset should be traumatic injury (caused by a single identifiable event) or gradual onset injury (caused by repeated microtrauma without a single identifiable event). Injuries can be further categorised into acute or chronic according to duration, whereby an acute injury is typically defined as one that is resolves within a 3-month period whilst a chronic injury lasts more than 3 months (*Jordan et al., 2010*). There is a paucity of epidemiological research specific to surfing using the injury definition of both injury onset (traumatic or gradual) and duration (acute or chronic), with only one study that has appropriately used these definitions, however this study focussed on gradual onset injuries only (*Remnant et al., 2020*).

Given the high prevalence of injury associated with surfing, the need for research into surfing related traumatic injuries that encompasses injuries of acute and chronic duration, and the lack of research in NZ despite its popularity there, the primary purpose of this study was to examine traumatic surfing injuries in NZ specific to injury incidence, duration, location, type, mechanism of injury and associated risk factors.

## METHODS

### Procedure

Ethical approval was granted by the Unitec Research Ethics Committee (2015-1032). A retrospective cross-sectional questionnaire was used to collect epidemiological data on surfing injuries from surfers *via* an online platform (Survey Monkey, Palo Alto, CA, USA). The questionnaire was piloted by twenty participants before being actively promoted from December 29th, 2015 to July 2nd, 2016. A dual recruitment process was used to promote the survey; the majority of participants (>95%) accessing the questionnaire online and a small minority of participants completed the questionnaire through face-to-face interactions with a researcher at popular surf locations in NZ. News articles, advertisements on a popular surf report site, promotion through surfing related Facebook groups and community noticeboard pages, and paid Facebook advertisements on a specifically created Facebook community page for this study were used as media promotion of the questionnaire.
## Participants

Participants who self-identified as "surfers currently in New Zealand" including both residents and those surfers visiting NZ and were aged 8 years or older were invited to complete the questionnaire. Parental or caregiver consent and supervision was required for participants aged under 16 years. Written consent was required for all participants. Surfing stance (either natural or goofy) and type of surfboard predominantly used (either short-board, mini-mal, long-board, or equal combination of more than one type of board) were used to further establish participants as surfers. Only participants who had been an active surfer for at least 12 months were included in the data analysis, similar to previous research (Furness et al., 2015).

Assuming an estimated proportion of injuries of 50% and a confidence level of 95%, a minimum of 1,068 respondents was calculated to be required to achieve a margin of error of three percentage points based on the 2014 estimated population of 155,000 surfers in NZ (Haughey, Gray & Heffield, 2015; Remnant et al., 2020). This population estimate was also used by Remnant et al. (2020) who investigated gradual onset injuries from the same study.

## Questionnaire design

The questionnaire was modified from a previous survey of Australian surfers (Furness et al., 2014; Furness et al., 2015) and structured in two sections. Section one included demographic questions determining current residence in NZ (a minimum of a 6-month stay), gender, age, ethnicity (allowing selection of more than one), years of surfing, participation type (recreational or competitive), time spent surfing (divided into summer and winter seasons), and surfing locations specific to NZ. An adapted version of the Hutt scale of surfing levels, which includes estimations of peel angle and wave height that can be successfully surfed, was also used to improve clarity regarding surfing ability (Hutt, Black & Mead, 2001). Surfing ability was further categorised during data analysis as either 'beginner' (including 'absolute beginner' and 'beginner'), or 'experienced' (including 'intermediate', 'advanced' and 'expert'). During data analysis board use was also further categorised as either 'predominantly short board' (including 'short board' and 'equally short board & mini-mal') or 'predominantly long board' (including 'mini-mal', 'long board (9 ft plus)' and 'equally mini-mal & long board'). There was an additional category that participants could select titled 'equally long and short board' which was not included in the above categories and filtered from the analysis. Competitive status was categorised as either; 'recreational' (including surfers who never competed in an event), or 'competitive' (including surfers who previously or currently competed in either a local, national or international event). Wave size was categorised as either; 'head height or smaller', or 'overhead or greater in height'. Aerialist manoeuvre capability was categorised as either; 'aerialist' (which included surfers who were able to propel themselves and the board in the air and land back on the water standing on the board) or non-aerialist.

Section two included questions about surfing-related injuries experienced in the preceding 12 months. Questions were specific to onset of injury, location, type, mechanism, diagnosis from a health professional, duration of injury, and subsequent management and

treatments. Respondents were required to identify injuries as being either 'traumatic' or 'gradual' using descriptions adapted from the definitions of *Verhagen & Van Mechelen (2010)*. Traumatic injuries were defined as having a specific event or sudden impact that occurred while surfing just prior to any symptoms (*i.e.,* pain) (*Verhagen & Van Mechelen, 2010*). For the purpose of this study, only injuries identified as 'traumatic' were analysed, all data pertaining to gradual onset injuries were previously published by *Remnant et al. (2020)*. Traumatic injuries were further categorised as either 'major' or 'minor' using a method previously employed by surf injury research (*Furness et al., 2014*; *Furness et al., 2015*). Major injuries were defined as those seeking treatment from a health professional, and/or at least one day off surfing and/or at least one day off work. Treatment from a health professional represented any form of service actively sought by participant due to the severity of the injury, including medical services, physiotherapy, osteopathy, chiropractic, naturopathy, specialist and/or practitioners of traditional Māori healing, traditional Chinese medicine, and acupuncture. Minor injuries were defined as those that did not require time off work or surfing or require treatment. Only injuries categorised as 'major' were analysed and presented.

To appropriately classify location of injury each body region was further divided into 10 body parts, using a condensed version of the Orchard Sports Injury Classification System (*Fuller et al., 2006*; *Verhagen & Van Mechelen, 2010*). The design of this study allowed participants to document more than one injury at each body location. Injury duration was either acute, defined as injury timeframe less than 3 months or chronic duration defined as injury recovery or ongoing timeframe greater than 3 months. The mechanism of injury was the movement that occurred just before or contributed to the injury. Mechanism of injury was categorised into three key areas which included direct contact (for example with the surfer's own board), approaching or riding the wave (for example during paddling) or other (for example through contact with marine life). Injury type was categorised into five groups (skin, bone, joint and ligament, muscle and tendon, and nerve). These categories for both injury mechanism and type were the same definitions used by *Furness et al. (2015)*. The questionnaire also allowed participants to document whether they were given a diagnosis from a health professional for the shoulder and knee only.

## Injury definitions

*Incidence.* Risk and rates of injury are two measures of injury incidence, and have been previously defined by *Knowles, Marshall & Guskiewicz (2006)*, where incidence proportion (IP) is the probability of an athlete getting injured over a 12-month period. To calculate IP, the number of surfers injured was divided by the total number of surfers exposed to risk (total injured surfers/total surfers). Incidence rate (IR) was defined as the incidence of injury over 1000 h of surfing (total number of injuries/1000 h surfing).

## Data analysis

Frequencies and descriptive statistics were used to summarise variables specific to injury location, injury type, and mechanisms of injury, participant ethnicity, and geographical regions commonly surfed. Chi-square test and relative risk (RR) were used to determine

**Table 1  Participant physiological and surfing demographics.**

|  | Total ($n = 1,473$) | Males ($n = 1,204$) | Females ($n = 269$) |
|---|---|---|---|
| Physiological demographics, mean ± SD |  |  |  |
| Age, y | 34.6 ± 11.9 | 35.3 ± 12.3 | 31.4 ± 9.3 |
| Weight, kg | 77.9 ± 14.0 | 81.3 ± 12.5 | 62.5 ± 8.8 |
| Height, m | 1.77 ± 0.1 | 1.80 ± 0.1 | 1.67 ± 0.1 |
| Body mass index, kg/m$^2$ | 24.8 ± 4.4 | 25.3 ± 4.3 | 22.7 ± 4.3 |
| Surfing demographics |  |  |  |
| Time surfing summer, h, mean ± SD | 147.2 ± 166.4 | 152.14 ± 167.0 | 125.25 ± 162.5 |
| Time surfing winter, h, mean ± SD | 66.9 ± 109.7 | 70.7 ± 111.5 | 50.1 ± 99.9 |
| Time surfing, h/y, mean ± SD | 214.1 ± 251.2 | 222.8 ± 251.2 | 175.3 ± 248.1 |
| Competitive involvement, n (%) | 542 (36.8) | 466 (38.7) | 76 (28.3) |
| Aerialist, n (%) | 206 (14.0) | 202 (16.8) | 4 (1.5) |

differences for categorical data and injury counts. Alpha was set at $P < 0.05$ *a priori* to determine significance. IR and IP were calculated for competitive status and aerialist manoeuvre capability. SPSS statistics package (IBM SPSS Statistics 27.0, IBM Corp., Armonk, NY) was used to analyse the data.

# RESULTS

## Participation and physiological demographics

After removal of 300 questionnaires due to incomplete key variables of injury and/or surfing experience, 1,542 respondents completed the full questionnaire. After excluding another 69 respondents who had <12 months surfing experience, 1,473 respondents were included in the final data analysis. The minimum and maximum ages were 8 to 74 years respectively with an interquartile range of 26 to 42 years. No differences were identified in the number of injuries between male and females (34.0% *versus* 34.6%, $\chi^2(1473) = 0.036$, $P = 0.85$, respectively). Table 1 presents participants' physiological and surfing demographics.

Participants of thirty-one ethnicities participated in this study, with the most selected ethnicity being NZ-European ($n = 1259$, 85.5%) and/or NZ-Maori ($n = 181$, 12.3%) as participants were allowed to select multiple ethnicities. Participants could also select the region they predominantly surfed within NZ. Figure 1 represents the dispersion of these regions with Auckland (30.9%) and Waikato (14.7%) having the highest frequencies.

## Incidence rate (IR) and Incidence proportion (IP)

Of the 1,473 eligible respondents, 502 (34.1%) reported sustaining a major traumatic injury. From these injured surfers, a total of 702 major traumatic injuries were reported as respondents were able to report up to two injuries per body part. The overall IP and IR were calculated to be 0.34 (0.32–0.37) and 2.23 (2.06–2.39) respectively; a breakdown of both IR and IP for the various cohorts is presented in Table 2.

## Injury location

*Location.* The highest frequency of injuries reported was to the head and face for the entire cohort, recreational, competitive and aerialist surfers (23.6%, 26.8%, 19.9% and 21.8%,

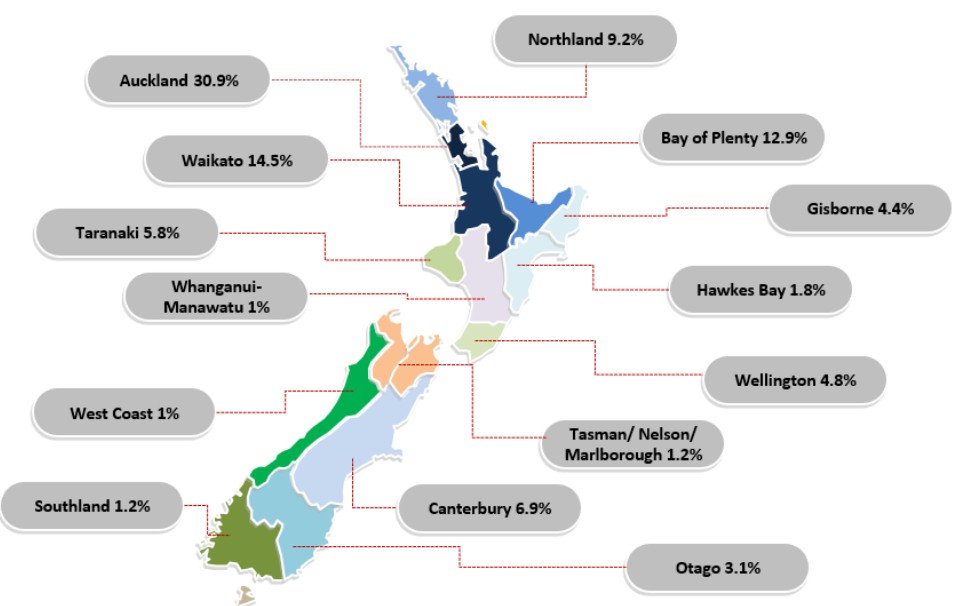

**Figure 1** **Distribution of primary regions surfed within New Zealand.** Note: A category of 'Other' was used which represented 1.4% of responses as the exact location could not be determined or was outside of NZ.

**Table 2** Major traumatic injury incidence proportion and incidence rate by competitive involvement, ability to perform aerials and overall.

| | Number of injured surfers (n) | Total surfers (n) | Incidence Proportion: *Total injured surfers/Total surfers*[a] | Cumulative number of injuries[b] | Cumulative Hours surfed per year | Incidence Rate: *Total number of Injuries/1000 h surfed* |
|---|---|---|---|---|---|---|
| Recreational | 280 | 931 | 0.30 (0.27–0.33) | 380 | 163,222 | 2.33 (2.09–2.56) |
| Competitive | 222 | 542 | 0.41 (0.37–0.45) | 322 | 152,194 | 2.11 (1.88–2.35) |
| **Total[c]** | **502** | **1,473** | **0.34 (0.32–0.37)** | **702** | **315,416** | **2.23 (2.06–2.39)** |
| Aerialist | 99 | 206 | 0.48 (0.41–0.54) | 151 | 76,138 | 1.98 (1.67–2.30) |

**Notes.**
[a]Values in parentheses are 95% CIs.
[b]As each surfer could report on more than one injury at the same site this value refers to the total number of injuries
[c]Total refers to the sum of the number of recreational and competitive surfers combined as aerialist were both recreational and competitive surfers.

respectively). Other commonly reported regions were ankle and neck for recreational surfers (13.2% and 10.8% respectively); neck and shoulder for competitive surfers (13% and 12.7%) and knee and shoulder for aerialists (15.2% and 13.9% respectively). Table 3 provides the frequency counts for all body locations.

When comparing the IP for individual body locations, a higher proportion of competitive compared to recreational surfers sustained injuries at the neck (6.8% *vs* 4%, $\chi^2$ (1,1473)

**Table 3  Location and total number of major traumatic injuries for surfers in New Zealand.**

| Site | Total Injuries[a] | | Recreational | Competitive | Aerialist Injuries n (%) |
|---|---|---|---|---|---|
| | n | % | n (%) | n (%) | |
| Head Face | 166 | 23.6 | 102 (26.8) | 64 (19.9) | 33 (21.8) |
| Neck | 83 | 11.8 | 41 (10.8) | 42 (13.0) | 12 (7.9) |
| Shoulder | 81 | 11.5 | 40 (10.5) | 41 (12.7) | 21 (13.9) |
| Arm | 36 | 5.1 | 23 (6.1) | 13 (4.0) | 10 (6.6) |
| Rib sternum | 41 | 5.8 | 24 (6.3) | 17 (5.3) | 7 (4.6) |
| Upper back | 22 | 3.1 | 9 (2.4) | 13 (4.0) | 5 (3.3) |
| Low back | 71 | 10.1 | 31 (8.2) | 40 (12.4) | 15 (9.9) |
| Hip/Groin | 47 | 6.7 | 24 (6.3) | 23 (7.1) | 12 (7.9) |
| Knee | 76 | 10.8 | 36 (9.5) | 40 (12.4) | 23 (15.2) |
| Lower leg/Ankle | 79 | 11.3 | 50 (13.2) | 29 (9.0) | 13 (8.6) |
| Total | 702 | 100 | 380 (100) | 322 (100) | 151 (100) |

**Notes.**
[a]Total refers to the sum of the number of recreational and competitive surfers combined as aerialist were both recreational and competitive.

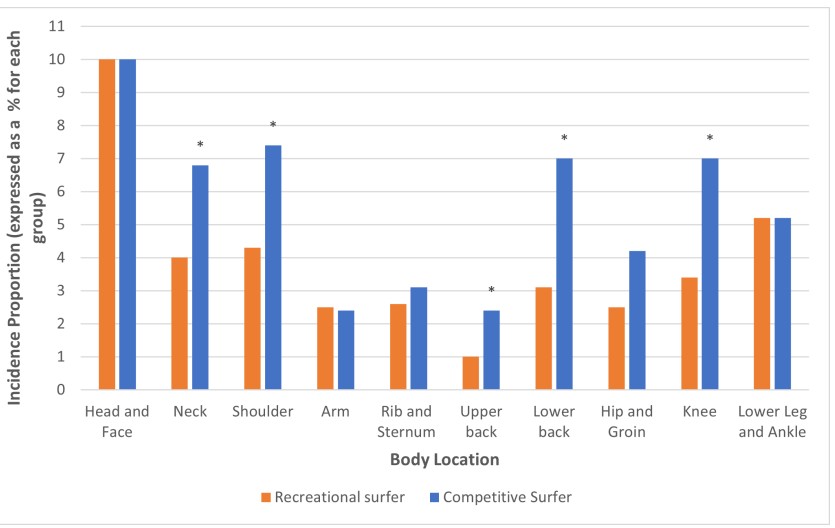

**Figure 2  Comparison of incidence proportion between competitive and recreational surfers at 10 body locations.** *Indicates statistical significance ($P \leq 0.05$).

$= 5.84$, $P = 0.019$); shoulder (7.4% *vs* 4.3%, $\chi^2$ (1,1473) $= 6.34$, $P = 0.017$), upper back (1% *vs* 2.4%, $\chi^2$ (1,1473) $= 4.77$, $P = 0.043$), lower back (7% *vs* 3.1%, $\chi^2$ (1,1473) $= 11.98$, $P = 0.001$) and knee (7% *vs* 3.4%, $\chi^2$ (1,1473) $= 9.67$, $P = 0.003$). Figure 2 presents proportional differences between recreational and competitive surfers.

The IP was significantly higher for surfers who had the capacity to perform aerial manoeuvres, with a significantly higher proportion of knee injuries compared to those surfers who are unable to perform aerial manoeuvres (9.7% *vs* 3.9%, $\chi^2$ (1,1473) $= 13.00$, $P = 0.001$).
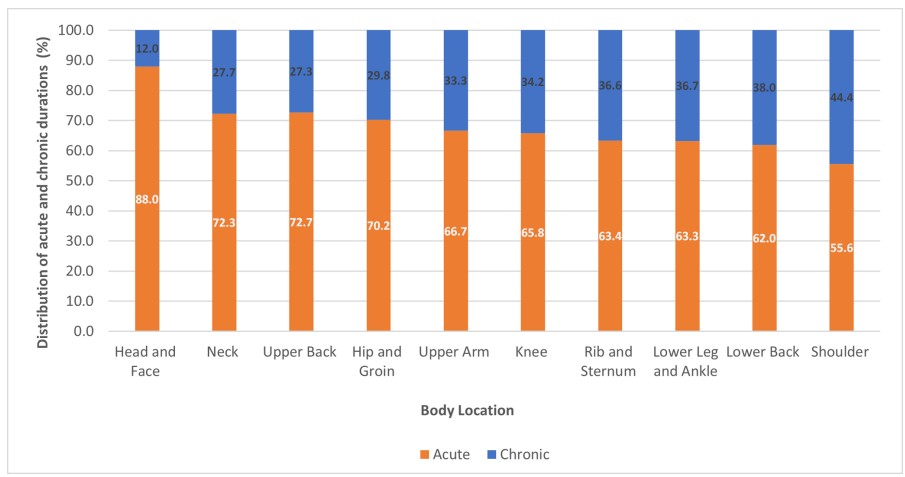

**Figure 3 Comparison of acute and chronic injury duration major traumatic injuries by body location.**

*Duration:* Shoulder injuries made up the largest proportion of chronic duration traumatic injuries (44.4%) whereas the head and face revealed the largest proportion of acute duration injuries (88%), see Fig. 3 for further details.

*Type*: Muscle and tendon injuries were reported most frequently (25.6%), followed by joint and ligament (23.9%) and skin (20.8%). Bone (9.9%) and nerve injuries (7.8%) occurred less frequently. Nervous system injuries such as brain injury or concussion (3.5%), eye (3%) and ear injuries such as a burst or perforated eardrum (1.5%) were the least common injury types and were only applicable to the head/face region. Further injury type and body location details are presented in File S1.

*Mechanism*: Direct contact injuries accounted for 58.1% of all injury mechanisms: approaching or riding the wave accounted for 35.3%, with the remaining 6.6% as other (*e.g.*, sustained from marine life). When specifically analysing direct contact mechanisms, the surfer being struck by their own board accounted for 44.6% of these injuries, followed by striking the seafloor (23.5%) and striking the surface of the sea (21.1%). When specifically analysing mechanisms associated with approaching or riding the wave the most common points of injury were during the take-off (16.1%), whilst duck diving (11.3%) and during a floater manoeuvre (10.5%). A detailed breakdown of all mechanisms of injury are presented in Table 4.

*Diagnosis.* A total of 135 of 702 (19.2%) major traumatic injuries were given a formal diagnosis. In the shoulder ($n = 77$), rotator cuff muscle tears were the predominant diagnosis (41.6%), followed by ligament sprains (22.1%) and subluxation/dislocation (22.1%). A total of 58 major knee injuries were diagnosed. Cartilage or meniscus injury encompassed 43.1% of knee injuries diagnosed, followed by ligament sprains, 41.4%. Other diagnoses included dislocation (6.9%) and other (8.6%). Details of knee and shoulder injury diagnosis by health care professionals are presented in Table 5.

**Table 4  Mechanism and site of major traumatic injuries.**

| Mechanism of Injury | Head/ Face | Neck | Shoulder | Upper back | Site of Injury Rib/ Sternum | Arm | Low back | Hip/ Groin | Knee | Ankle/ Lower leg | Total, n (%) |
|---|---|---|---|---|---|---|---|---|---|---|---|
| **Direct contact injuries** | | | | | | | | | | | |
| Struck (own board) | 93 | 11 | 7 | 2 | 18 | 13 | 3 | 5 | 6 | 24 | 182 (25.9) |
| Struck (other board) | 16 | 2 | 3 | 1 | 1 | 1 | 7 | 3 | 2 | 8 | 44 (6.3) |
| Striking seafloor | 18 | 23 | 11 | 4 | 3 | 8 | 6 | – | 5 | 18 | 96 (13.7) |
| Striking sea surface | 17 | 27 | 19 | 6 | 6 | – | 5 | 3 | 1 | 2 | 86 (12.3) |
| *Total, n* | *144* | *63* | *40* | *13* | *28* | *22* | *21* | *11* | *14* | *52* | *408 (58.1)* |
| **Approach/wave riding** | | | | | | | | | | | |
| Paddling | 1 | 1 | 8 | 1 | – | 1 | 3 | 1 | – | – | 16 (2.3) |
| Duck diving | 3 | 4 | 14 | 2 | 1 | 1 | 2 | – | 1 | – | 28 (4) |
| Take off | 3 | 4 | 2 | 2 | 1 | 1 | 9 | 6 | 7 | 5 | 40 (5.7) |
| Bottom turn | – | – | – | – | – | – | 4 | 4 | 4 | 1 | 13 (1.9) |
| Top turn | – | 1 | 2 | 1 | – | – | 4 | 5 | 9 | 3 | 25 (3.6) |
| Cut back | – | – | – | – | – | – | 5 | 3 | 9 | 1 | 18 (2.6) |
| Re-entry | 1 | – | 1 | – | 3 | – | 6 | 4 | 5 | 4 | 24 (3.4) |
| Floater | 3 | 2 | 3 | 1 | 1 | – | 3 | 2 | 7 | 4 | 26 (3.7) |
| Aerial | 1 | – | 1 | 1 | – | 1 | 1 | 1 | 4 | 2 | 12 (1.7) |
| Riding wave face | 1 | 2 | 1 | – | 1 | 1 | – | 2 | 4 | 1 | 13 (1.9) |
| Tube riding | 5 | 3 | 3 | – | 2 | – | 4 | 2 | 3 | 1 | 23 (3.3) |
| Wave force | 1 | 2 | 3 | – | – | – | 2 | 1 | 1 | – | 10 (1.4) |
| *Total, n* | *19* | *19* | *38* | *8* | *9* | *5* | *43* | *31* | *54* | *22* | *248 (35.3)* |
| **Other** | | | | | | | | | | | |
| Marine life encounter | 3 | – | – | – | – | 2 | – | – | – | 1 | 6 (0.9) |
| Other* | – | 1 | 3 | 1 | 4 | 7 | 7 | 5 | 8 | 4 | 40 (5.7) |
| *Total, n* | *3* | *1* | *3* | *1* | *4* | *9* | *7* | *5* | *8* | *5* | *46 (6.6)* |
| **Total, n** | **166** | **83** | **81** | **22** | **41** | **36** | **71** | **47** | **76** | **79** | **702 (100)** |

Notes.
*Where possible the text provided was re-coded into the pre-specified mechanisms (such as Paddling etc.), however in the cases where this was not possibe due to the vast differences in the responses the mechanism remained as 'Other'.

## Risk factors

The relative risk ratios were identified for several variables within this study. Intermediate or above skill level surfers were more likely to sustain an injury compared to beginner surfers (35.8% *vs* 22.7%, RR = 1.58, $P < 0.001$). Competitive surfers (41% *vs* 30.1%, RR = 1.36, $P < 0.001$) and surfers with the ability to complete aerials sustained significantly more injuries than surfers who were not (48.1% *vs* 31.8%, RR 1.51, $P < 0.001$). When analysing the short board *versus* long board categories participants who indicated that they rode both a long and short board equally ($n = 132$) were excluded in the analysis. Predominantly riding a short board ($n = 1003$) verses predominantly riding a long board ($n = 338$) increased injury risk (35.4% *vs* 29.6%; RR = 1.196, $P = 0.051$), however this was not statistically significant.

An additional sub-analysis was conducted to determine the risk of sustaining an injury if the surfer could perform an aerial with the data set filtered to those surfers who rode a

**Table 5  Injury diagnosis provided by health professional for the shoulder and knee.**

| Location | Diagnosis | Injury 1, n | Injury 2, n | Total, n (%) |
|---|---|---|---|---|
| Shoulder | Ligament sprain | 16 | 1 | 17 (22.1) |
| | Dislocation/Subluxation | 16 | 1 | 17 (22.1) |
| | Cartilage | 3 | 0 | 3 (3.9) |
| | Rotator cuff | 32 | 0 | 32 (41.6) |
| | AC joint | 6 | 0 | 6 (7.8) |
| | Unknown/Other | 2 | 0 | 2 (2.6) |
| | Total | 75 | 2 | 77 (100) |
| Knee | Ligament sprain | 23 | 1 | 24 (41.4) |
| | Cartilage/Meniscus | 22 | 3 | 25 (43.1) |
| | Dislocation | 3 | 1 | 4 (6.9) |
| | Unknown/Other | 5 | 0 | 5 (8.6) |
| | Total | 53 | 5 | 58 (100) |

short board. This was conducted based on the assumption that a surfer riding a long board or mini mal could not complete an aerial. All participants who reported riding a long board or a mini mal or a combination of these two were excluded in the analysis. Those surfers who reported riding a short board ($n = 919$), a combination of minimal and short board ($n = 137$) or a combination of a short board and long board ($n = 132$) were included in the total data set of 1,135 surfers. The relative risk ratio was then calculated with those surfers who had an ability to complete aerials sustaining significantly more injuries (49.0% *versus* 32.6%, RR 1.50, $P < 0.001$).

## DISCUSSION

The primary purpose of this study was to examine traumatic surfing injuries specific to injury incidence, duration, location, type, mechanism of injury and associated risk factors in NZ surfers. Given that the total participant numbers exceeded the required estimated sample size calculation, this study provides a sample representative of the NZ surfing population. With over 1,400 participants, this study is likely to provide a sample representative of the broad recreational and competitive NZ surfing population. Given the lack of surfing injury epidemiology research in NZ, the large participation rates in the country, and the high injury claim counts, this large cross-sectional study provides much needed data. Consequently, these results may reflect the range of injuries that are likely to be sustained globally as surfing increases in popularity.

### Incidence rate, incidence proportion and relative risks

This study revealed that the ability to perform an aerial manoeuvre resulted in the highest IP (0.48), followed by competitive status (IP 0.48) and recreational (IP 0.30) populations. When examining incidence rate (IR) measured by the number of injuries per 1000 h surfed, the results were reversed where greater rates were seen in recreational surfers (IR 2.33) compared with aerialists (1.98). This reversed trend is explained as aerialists and competitive surfers had higher participation rates (hours per year) compared to recreational surfers,

which in turn reduced the calculated IR. In summary, nearly one in every two aerialist and competitive surfers sustained an injury in the last year as opposed to one in every three recreational surfers, despite both groups having a lower IR (injuries per 1000 h) compared to the recreational surfers. Similar results were reported by *Furness et al. (2015)* with aerialists showing a greater IP (0.48) than competitive (0.42) and recreational (0.35) surfers, and recreational surfers showing a higher IR (2.18) than competitive (1.51) and aerialist (1.35) surfers. *Nathanson et al. (2007)* conducted a prospective study analysing all acute surfing injuries across 10 amateur and 22 professional competitions globally, from 1999 to 2005. The calculated IR was four times higher than the current study at 6.6 significant injuries per 1000 h of competitive surfing. Similarly, *Inada et al. (2018)* reported an IR of 6.6 injuries per 1000 h of competitive surfing across 50 contests during the Japan Pro Surfing Tour between 2009 and 2016. A possible factor as to why both studies (*Inada et al., 2018*; *Nathanson et al., 2007*) had higher IRs compared to the current study is that both studies collected data prospectively at the time of the injury. Data collected retrospectively runs the risk of memory decay and it is highly likely that injuries are missed and or estimated hours surfed are over or underestimated thus misrepresenting the true IR and IP (*Jenkins et al., 2002*). It also needs to be highlighted that both studies used a professional cohort competing at a high level and thus the physiological demands are likely to be higher than their recreational counterparts potentially predisposing them to greater injury risk.

The highest relative risk ratio was seen in surfers who had the ability to perform an aerial, compared to those who could not perform this manoeuvre. This could be attributed to the higher loads absorbed through the lower limbs and awkward knee and ankle joint positions when landing these manoeuvres (*Lundgren et al., 2014*; *Lundgren et al., 2016*) resulting in an increased likelihood of injury (*Furness et al., 2015*; *Lundgren et al., 2014*). In addition to the kinematic challenges of the aerial, the amount of vertical force estimated during aerial manoeuvres is almost six times a surfers body weight, thus providing substantial stress within the lower body (*Lundgren et al., 2016*). While this rationale for a higher injury count in this cohort appears sound, it is not supported by the mechanism of injury data within this current study, which revealed only 1.7% of all mechanisms being attributed to aerials. It could be postulated that aerial manoeuvres are complex and difficult to execute (*Forsyth et al., 2021*) requiring more experience and practice, therefore exposing the surfer to greater risk through more hours and years of surfing compared to other surfers such as beginners who are not performing these manoeuvres. Another possible theory is that surfers who can complete aerials engage in more risk-taking manoeuvres and consequently have higher injury counts.

Higher relative risk ratios in competitive surfing could be attributed to the greater physiological demands of competitive surfing compared with typical recreational surfing sessions (*Farley, Abbiss & Sheppard, 2017*). Furthermore, the scoring of professional competitive surfing is based on wave difficulty, variety and combinations of major manoeuvres, speed, power and flow on the wave (*World Surf League, 2021*), therefore requiring competitive surfers to push themselves physically further in training than a recreational surfer may do, so increasing the likelihood of injury.

## Injury location, duration, type and mechanism

Injuries to the head and face had the highest injury count (23.6%), with neck injuries second (11.8%), compared to all other locations. The current study results are similar to a recent systematic review by *McArthur et al. (2020)* who reported face, head and neck injuries represented 33.8% of total acute injuries. Nine of 19 included studies reported the head, face and neck to be the most common body regions affected. Head, neck, and eyes (31.7%) were also identified as the most common locations of injury by *Moran & Webber (2013)* in their examination of surfing injuries requiring first aid at NZ beaches.

When analysing injury locations between competitive and recreational surfers, several joints had significantly higher proportions of injuries, specifically the shoulder, lower back, and knee (see Fig. 2). Higher proportions of injuries in these locations could be explained by demands placed on these surfers previously discussed above. When looking at aerialists, the knee had the second highest injury count (15.2%). A possible theory that links aerial manoeuvres and knee injury; is that successful completion of a aerial manoeuvre commonly involves placing the knee in a valgus position when landing (*Forsyth et al., 2018*). The knee joint may compensate for range of motion restrictions at the ankle and hip and be predisposed to injury (*Devita & Skelly, 1992*).

In addition to determining injury location, this study also presented results on duration of injury. The head and face appear to have the highest proportion of acute durations (less than 3 months) as opposed to the lower back (38%), lower leg and ankle (36.7%) and shoulder (44.4%) which revealed high frequencies of chronic durations (greater than 3 months). This high proportion of chronic shoulder injury durations is an important finding as it presents the potential difficulties of surfers recovering from an injury at this location shoulder. *Remnant et al. (2020)* who investigated gradual onset injuries in surfers also revealed high injury counts at the shoulder of a chronic duration. There are several potential reasons as to why the shoulder is commonly associated with chronic injury. Firstly, 50% the time spent surfing involves paddling (*Farley, Abbiss & Sheppard, 2017*); to put this in perspective, a two-hour recreational surf session will involve approximately 60 min of paddling and covering a distance of 5 km (*Farley, Abbiss & Sheppard, 2017*). This external load placed on the shoulder joint may predispose the shoulder to injury (specifically the rotator cuff). Increased training and competition volume and arm fatigue has been shown to be a risk factor of chronic shoulder injury in overhead sports such as baseball (*Norton et al., 2019*) and swimming (*Khodaee et al., 2016*). It could be further postulated that many of the surfers with a shoulder injury continue to participate in surfing despite symptoms or return to surfing despite inadequate recovery. This has previously been the case in the sport of swimming (*Beach, Whitney & Dickoff-Hoffman, 1992*). In addition, many surfers may have underlying modifiable risk factors (such as range of motion deficits) which predisposes them to shoulder injuries of a chronic duration. This hypothesis has further been confirmed by *Clarsen et al. (2014)* who investigated handball players and demonstrated that reduced glenohumeral rotation, external rotation weakness and scapular dyskinesis were risk factors to chronic shoulder injury. In addition, swimmers with either high (100 degrees or greater) or low (less than 93 degrees) external rotation range of motion have been shown to be at significantly higher risk of developing shoulder

pain (*Walker et al., 2012*). Lastly, the complexity of the shoulder joint and rehabilitation must not be neglected when postulating potential reasons for higher proportions of chronic shoulder injury (*Cools et al., 2020*). Traditionally, rehabilitation programs are designed to improve flexibility, strength, endurance, functional stability and motor control (*Cools et al., 2020*). Trying to address all of these issues, in addition to challenges with patient compliance provide further rationale for the shoulder having the highest proportion of chronic injury durations. Data specific to injury duration is unique to this study as previous research has often failed to address the nature of injury (traumatic or gradual onset) and the duration (acute or chronic) separately. Specifically the recent systematic review by *McArthur et al. (2020)* included studies that investigated injuries occurring from a specific event that lasted less than 3 months. Since this study reported almost a third of traumatic injuries are of chronic duration having a recovery period of greater than 3 months, previous studies potentially underestimate the prevalence of traumatic injuries, thus highlighting the strength of the current study.

### Injury type

When addressing injury type, muscle and tendon injuries were the most common (25.6%), followed by joint and ligament injuries (23.9%). Similarly *Furness et al. (2015)* identified muscular (31.3%) and joint injuries (28.7%) to be the most common types of injuries in Australian surfers. In contrast, previous research by *Moran & Webber (2013)* found lacerations and abrasions to be the most common nature of injury (59.2%), however, this study lacked a category for sprains and strains in injury reporting. In that study bruising represented the second most common nature of injury (15.2%) and could have included sprains or strains but would be indistinguishable from simple contusions. *McArthur et al. (2020)* found skin injuries to be the most common type of injury overall (46.0%), followed by soft tissue injuries (22.6%). Differences in injury type frequencies between studies could be explained by the definition and categories used and the location of where the data was collected. For example, data collected out of emergency departments (*McArthur et al., 2020*) or specific to first aid (*Moran & Webber, 2013*) may tend to have a higher representation of skin injuries as opposed to survey based data collection methods similar to the current study (*Furness et al., 2015*). Furthermore, increased manoeuvrability of modern boards, combined with a higher risk-taking propensity of surfers such as completing aerials and manoeuvres in the critical and most powerful parts of the wave may have led to an increase in muscle and joint related injuries (*Furness et al., 2015*).

The current study also allowed for participants to document a diagnosed injury type by a health professional at the knee and shoulder. The high frequency of cartilage and menisci (43.1%) and ligamentous (41.4%) injuries of the knee within the current study is supported by *Hohn et al. (2018)* who reported the most common knee diagnoses in professional surfers were medial collateral ligament (MCL) sprains (49%) and meniscus tears (37%). Similarly, *Inada et al. (2018)* reported the majority (43.5%) of knee injuries in competitive surfers during competition were the MCL. These results may be explained by the large compressive loading forces experienced at the knee with aggressive turning movements required in sections of the wave and during manoeuvres such as floaters and
aerials. Previous research has also described that MCL injuries tend to occur in surfers' back knees, caused by high compressive valgus forces on landing aerial manoeuvres (*Burgess, Swain & Lystad, 2018*; *Hohn et al., 2018*).

The most common diagnosis at the shoulder was rotator cuff pathology (41.6%). These results are not surprising given the paddling requirements of surfing involving several shoulder movements all of which involve active contraction of the rotator cuff. While this causal effect seems probable, very few studies have specifically investigated shoulder pathology in surfing populations (*Langenberg et al., 2021*). The findings of the current study are supported by a case series of 25 surfers which revealed that 31% of the surfers had rotator cuff tendonitis (*McBride & Fisher, 2012*). *Remnant et al. (2020)* investigated gradual onset injuries and found the rotator cuff injury diagnosis was the highest of all shoulder diagnoses within surfers in NZ. Given the high frequency of rotator cuff pathology further research is needed in understanding the pathogenesis of these pathologies to enable appropriate intervention.

## Injury mechanism

Direct contact injuries (58.1%) were the most common category of injury mechanisms in this study. Surfers being struck by their own boards was the most common mechanism of injury, representing the majority (44.6%) of direct contact injuries. A systematic review which investigated acute and traumatic injuries also reported that being struck by a surfers own board (38.6%) was the most common mechanism of injury overall (*McArthur et al., 2020*). Of all 19 included studies, 14 reported being struck by own board to be the most common mechanism of injury and four studies provided no data on injury mechanism. *Moran & Webber (2013)* who examined surfing injuries requiring first aid in NZ also reported being struck by the surfer's own board represented half (50.5%) of all reported injury mechanisms. The sharp rails and points of the modern resin boards, aimed at maximising performance may have exacerbated the capacity to inflict soft tissue trauma when impacting the surfer (*Dimmick et al., 2018*). It could be postulated that as surfing popularity increases, surf line-ups will inevitably become more crowded and the risk of trauma from another's surfboard may increase, from that seen in the current study where this represented only 6.3% ($n = 44$) of all injury mechanisms.

## Study limitations and strengths

There are limitations and strengths of this study that need to be explicitly outlined when generalising our findings beyond this study. Most notably is the design of the study being retrospective as opposed to using a prospective data collection method. The evident limitation with retrospective survey design is the inability to accurately recall the injury or the details specific to the injury (*Jenkins et al., 2002*). While a prospective design is optimal it is not necessarily feasible given the coastal location of surfers, the lack of one centralised data collection location and that many recreational surfers do not belong to a surf or board-riders club or other organisation where this type of data could be captured. To date, the only prospective studies have involved competitive surfers belonging to a professional body (*Inada et al., 2018*; *Nathanson et al., 2007*) where injury data is captured at the time

of injury. A further limitation of the study was the inability to provide details regarding injury management. The current study allowed for the respondent to document the type of practitioner (physiotherapist etc.) that attended for treatment, however no detail of the actual treatment that was provided. This study also used a very broad definition of a competitive surfer, which included current and previous competitive involvement and all levels of competitive surfing (such as local board rider clubs to those competing at an international level). Given the range of competitive levels caution should be applied when generalising results to all competitive surfers outside of this study.

A major strength of this study was the appropriate classification of injury onset by using the definitions of traumatic and gradual onset injuries. This further enabled injury durations to be appropriately classified as acute and chronic. Several previous surf specific epidemiological studies have not clearly delineated between injury onset and injury duration, limiting the understanding of these injuries (*Furness et al., 2015*; *Nathanson, Haynes & Galanis, 2002*; *Taylor et al., 2004*).

## Future research

This study provides a foundational understanding of injury epidemiology and answers the initial step in the injury prevention cycle described by *Van Mechelen, Hlobil & Kemper (1992)* of identifying the problem (step 1, injury registration). Also, this study begins to investigate the next step proposed by *Van Mechelen, Hlobil & Kemper (1992)* in the injury prevention cycle that aims to address the mechanism and causes of injury. Further research is needed to provide musculoskeletal profiles of various surfing cohorts to identify potential causes (*e.g.*, asymmetry in ROM or strength) and risk factors for sustaining an injury. For example, what is the normal shoulder strength profile of a surfer and do limitations or asymmetries in various movements result in higher incidence of shoulder injury? Alternatively, does hip and ankle range of motion influence knee injury prevalence? Prospective research is needed where profiles and assessments are conducted prior to injury to adequately determine injury risk factors. With respect to the head and face injuries, further research is required to evaluate the effectiveness of headgear in reducing the number and/or severity of head and face injuries in surfing before more thorough arguments can be made for their integration into surfing culture for safety.

Understanding mechanisms and risk factors of injury allows for a targeted prevention approach which highlights the third step in the injury prevention cycle (*Van Mechelen, Hlobil & Kemper, 1992*). In the final step of the injury prevention cycle the ability of a surf-specific strength and conditioning program to reduce injury incidence could be assessed (*Van Mechelen, Hlobil & Kemper, 1992*). Prevention programs have been implemented in numerous sports and are often not supported by clear evidence that injury is reduced through completing such a program (*Cools et al., 2020*). Ensuring higher quality research is completed will enable evidence-based programs to be implemented and assessed.

With the evidence outlined in the current study, prevention programs should look at initially screening surfers at higher risk (such as competitive and those completing aerials); assessing key injury locations for each specific cohorts (such as the shoulder and knee for those surfers completing aerials) for strength, range of motion and proprioception deficits.

This information can then be used to inform an individualized and appropriate strength and conditioning program.

## CONCLUSION

This study provides valuable information specific to surfing participation and injury epidemiology within NZ. One in every three recreational surfers sustained a traumatic injury and approximately one in every two competitive or aerialist surfers sustained a traumatic injury in a previous 12-month period. The head and face were the most common location of injury overall, with the shoulder having the highest proportion of injuries of a chronic duration, highlighting the long-standing recovery process at this location. Competitive surfers were more likely to sustain a neck, shoulder, upper back, lower back, and knee injury when compared to their recreational counterparts. The knee was a highly injured location for surfers completing aerial manoeuvres. Key risk factors for sustaining an injury included being an experienced surfer, performing aerial manoeuvres or competing in surfing competitions. Future research needs to build upon this knowledge and investigate mechanisms and causes of injury using musculoskeletal assessment and prospective injury research methods, thus enabling targeted injury prevention programs.

### Funding
The authors received a Todd Foundation scholarship, O'Neill sponsorship of a wetsuit for a participation prize draw, and participant recruitment advertising by Swell Map. The funders had no role in study design, data collection and analysis, decision to publish, or preparation of the manuscript.

### Grant Disclosures
The following grant information was disclosed by the authors:
Todd Foundation scholarship, O'Neill sponsorship of a wetsuit for a participation prize draw, and participant recruitment advertising by Swell Map.

### Competing Interests
Mike Climstein is an editor for PeerJ.

### Author Contributions
- James Furness conceived and designed the experiments, analyzed the data, prepared figures and/or tables, authored or reviewed drafts of the paper, and approved the final draft.
- Katherine McArthur and Darcy Jorgensen analyzed the data, prepared figures and/or tables, authored or reviewed drafts of the paper, and approved the final draft.
- Debbie Remnant, Catherine J. Bacon and Robert W. Moran conceived and designed the experiments, performed the experiments, authored or reviewed drafts of the paper, and approved the final draft.

- Wayne Hing and Mike Climstein conceived and designed the experiments, authored or reviewed drafts of the paper, and approved the final draft.

## Human Ethics

The following information was supplied relating to ethical approvals (i.e., approving body and any reference numbers):

The Unitec Research Ethics Committee granted ethical approval to carry out this study (ethics application number: 2015-1032).

## Data Availability

The raw data is available as a Supplemental File.

## Supplemental Information

Supplemental information for this article can be found online at http://dx.doi.org/10.7717/peerj.12334#supplemental-information.

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
