# Peer review of "Traumatic surfing injuries in New Zealand: a descriptive epidemiology study"

_PeerJ, doi:10.7717/peerj.12334_

## Round 0.1 · original submission · Major Revisions

Dear authors,

I have now received quick feedback from two experts, who both saw the merit in the article. They provided detailed reports for the article to be raised to the standard of publication. I have selected a "major" revisions as reviewers raise a number of valid points requiring clarification.

I would add to the review that I was expecting results and discussion regarding injury treatment (type, perceived efficacy) since this information seems to have been collected. Could the authors provide a rationale for not including this information that would add to the study?

Reviewer 1 ·

Basic reporting

Professional English language was used throughout the manuscript. The introduction an background lead nicely to the research aim of the study. Structure conforms to PeerJ standards. Figures are all relevant. Table 1 contains two times: Time surfing summer, h, mean etc. making it therefore impossible to understand what was meant by this. Please change accordingly.

Experimental design

The primary research is within the Scope of the journal. The research question is well defined, however, the relevance/justification is lacking on why surfing injuries need to be specifically investigated in NZ. In the manuscript it is only referred to that people within NZ were investigated. Where travellers, who have been in NZ at the time included? Please complete this information. The investigation was performed at a high technical & ethical standard and methods described with sufficient detail.

Validity of the findings

As mentioned above, the justification which specific knowledge gap this study will fill is missing some detail, as it is not quite clear what the novelty of the NZ specific data will add to the literature. The data set including more than n=1400 subjects is really strong, however, not quite clear in the introduction and later in the discussion what the data adds to the existing literature from other countries. Especially with regard to the systematic review from McArthur et al. (2020). A dataset of more than n=1400 participants has the potential to inform here very specifically and this strength should be used.

A meaningful discussion of the data and the outcomes could be extended with regards to older adults. The authors showed a mean age of 34.6±11.9 years. What was the age range included in the investigation? This might have an impact on the interpretation of the data as well, as the severity of injuries might increase with age as well as the recovery time. Also there was no detailed analysis comparing between men and women. A short outline of prevention strategies is missing and would strengthen the manuscript in the discussion.

Overall the underlying data are robust, statistically sound and controlled. The conclusions are well stated and linked to the original research question, but could be improved if the above-mentioned points would be added.

Reviewer 2 ·

Basic reporting

Thank you for the opportunity to review this manuscript that sought to determine traumatic injury detail for surfers in New Zealand during 2015 - 2016.

Whilst I found this a very interesting manuscript containing novel findings on which authors are to be congratulated, I have suggestions to improve clarity, context, grammatical precision etc. Please see below for details:
• Line 70- 71 – 1st usage of abbreviation for New Zealand come before first full term and defined abbreviation. Also some countries are abbreviated and some used in full – consistency please.
• Line 74 'combing' or 'combining'?
• Line 74 – ‘large proportion of NZers who participate in adventure sports’ – this is referenced - can you be more specific, than the vague statement of ‘large’?
• Line 82 – Within? versus ‘on’ or ‘at’?
• Line 83 ‘limits’ instead of ‘limiting’
• Line 119 – ‘completed’ rather than ‘completing’
• Line 269 – why was eye or ear injuries listed as nervous system only? For example, ‘surfers ear’ is a common surfing complaint, but is not neurological.
• Table 1 – ‘Time surfing summer’ repeated – error? Winter instead?
• Line 314 – ‘recreational’ rather than ‘recreation’
• Line 321 – ‘research’ not necessary
• Line 343 – what is the difference between physiological and physical demands?
• A point for authors to consider: Recreational surfing sessions are longer than competitive events – more time = more load? Additionally, all competitive suffers also free surf aka are recreational.
• Line 347 – comma after ‘do’
• Line 353 – comma after ‘head’
• Line 357 – 359 – doesn’t make sense as written – common injuries between competitive and recreational surfers, ‘locations’ instead of ‘location’?
• Line 362 – remove ‘the’ and replace ‘a’ with ‘an’
• Line 363 – remove double space after ‘The’.
• Line 378 – baseball and handball comparison – did authors come across any other swimming or paddling sport evidence, as those results would be more transferable / valid due to muscle patterning, usage, functional anatomy etc.
• Line 380 – should it be ‘inadequate’ instead of ‘adequate’. Also, full stop after ‘recovery’.
• Line 394-395 – no need for ‘type overall’
• Line 399 – remove ‘all together’
• 407 – How is higher risk taking judged? In relation to aerials? Needs to be clearer please.
• Line 426 – remove ‘the’
• Line 435 -437 rephrase sentence please for better understanding / grammar.
• Lines 448 – 462: Based on previous comments by the reviewer, the reviewer believes there are more limitations to acknowledge. If so, please rephrase the first sentence of this paragraph to include limitations other than external validity aspects.
• Line 461 – comma after ‘duration ‘
• Line 483 – comma after ‘step’
• Line 482 – 485 – cumbersome. Rephrase for efficiency and understanding.
• Line 496 – 498 – suggest separate sentences for shoulder and knee to improve understanding by reader.
• More crowded line ups lead to more injuries – perhaps reference the 44 surfers struck by other surfer’s boards to make this point resonate with reader.
• In table 4 – Under ‘Other’ category, there is an ‘other’ subcategory which resulted in more injuries than aerials etc – what did this consist of? More detail would be appreciated and may suggest further avenues of research.
• Figure 3 and line 369 - 371: Whilst shoulder is the most common chronic location – other locations do not lag far behind – could this also be an important finding and potentially be the focus of future research as well?

Experimental design

• What was the rational for setting a minimum age of 8 for inclusion into the study? Given there were 100 participants under the age of 18, do authors believe this may have influenced findings related to injury mechanism and length of healing time etc. given the coordination, musculoskeletal, physiological differences and other considerations when comparing children to adolescents to adults and older adults? Additionally, was increasing age itself looked at as a predictor of injury?
• Line 134 – how did researchers determine the estimated proportion of injuries to be 50%? As this is integral to determine the number of participants / sample size needed, please provide more detail.
• Lines 141 – 156: Section detailing how someone is classified as competitive vs recreational vs level of experience is confusing and seems to include multiple different methods that is not expanded on later, but level of experience is mentioned as a major finding. Please clarify. As a example, was an international contest simply classified as overseas or meant to represent WQS/WCT (elite) level competition? Please elaborate.
• The competitive status definition encompasses a broad range of event levels and therefore intensity / skill sets / formats (local contest to national to international) as well as a long timeframe (previously competing in a local event many years ago could conceivably qualify the surfer into this category even if since this single event they had been recreational). This would have affected IP, IR and RR calculations and findings. Seeing as this was an important finding in relation to injury location etc., do the authors think this definition of competitive was too broad? This also relates to lines 342 – 347 of the discussion where competitive scoring and competition heat intensity etc is used to justify findings, whilst that is conceivably not a major factor for people identifying as competitive given my comments above. Please revise / acknowledge as necessary.
• If I understand the raw data file correctly, over 500 of the responders were long boarders. In the vast majority of cases, long boarders cannot perform aerials – therefore more than a third of responders would be classed as non-aerialists ‘automatically’ and not due to ‘capacity / ability’ per se. As only 206 surfers of the remaining pool identified as aerialists, do the authors think the inclusion of various board types influenced the sub analysis of aerialists being more prone to injury (RR etc) or should this issue be listed as a limitation?

Validity of the findings

• Line 179 – 182: Head/face region includes eye, ear, brain, skin etc. Other regions do not contain as many organs / subcategories. Does this perhaps contribute to the finding of this region being the most common traumatic injury location? Does this issue warrant sub analysis?
• Lines 309-315 – aerialists are made up of both competitive and recreational surfers. This makes the direct comparison statement seem problematic as written in lines 309-311. Please revise. Additionally, IR for recreational surfers mention in text (2.5) is different from table 2 (2.33). Please correct.
• Whilst in Table 2 99 aerialists report 151 injuries, In table 4, aerials themselves account for only 12 injuries (1.7% of total). In absolute terms, this is the second lowest in category, whilst take-off and floaters are much higher. Of interest to the reviewer, duck diving seems to result in greater injuries as well. How does this data line up with your statement of potential to do aerials as a risk factor / major finding whilst these other factors are not mentioned in the discussion?
• A related point: in lines 331 – 341 the initial discussion in this paragraph seems to support the argument that aerials themselves are likely to cause injury though higher forces etc. Whilst this seems reasonable, this line of argument does not seem to be supported by the data of this study, as mentioned in the previous point which suggests that aerialists are at an increased risk of injury, but not necessarily whilst doing airs. Please revise accordingly to make this clearer (the second half of this paragraph does this to some extent).
• Line 328 – 330 - here authors point out a limitation of their methods as a possible explanation. Given the professional nature of most of the referenced previous research, did the authors of those papers point to any intensity or physiological load variables for example that may have partly explained their higher IR’s? Suggested that alternative explanations, other than data collection method limitations would improve the current paper.

Additional comments

No additional comments

---

## Round 0.2 · accepted · Accept

Thank you for providing detailed and thoughtful replies to the comments provided by the authors and myself. It's been a pleasure to oversee this process.

Reviewer 1 ·

Basic reporting

The authors have addressed all comments to my fullest satisfaction.

Experimental design

The authors have addressed all comments to my fullest satisfaction.

Validity of the findings

The authors have addressed all comments to my fullest satisfaction.

Reviewer 2 ·

Basic reporting

The authors are to be commended on their responses to reviewers and editors previous comments and for the associated changes and additions they have made to the revised manuscript. No further comments from this reviewer.

Experimental design

No comment

Validity of the findings

No comment

Additional comments

No comment